# Large Language Models as Recommendation Systems in Museums



**Georgios Trichopoulos** [1,*]🆔**, Markos Konstantakis** [1]🆔**, Georgios Alexandridis** [2]🆔 **and George Caridakis** [1]🆔

1   Department of Cultural Technology and Communication, University of the Aegean, University Hill, 81100 Mytilene, Greece; mkonstadakis@aegean.gr (M.K.); gcari@aegean.gr (G.C.)
2   Department of Digital Industry Technologies, National and Kapodistrian University of Athens, 34400 Psachna, Greece; gealexandri@uoa.gr
*   Correspondence: gtricho@aegean.gr

**Abstract:** This paper proposes the utilization of large language models as recommendation systems for museum visitors. Since the aforementioned models lack the notion of context, they cannot work with temporal information that is often present in recommendations for cultural environments (e.g., special exhibitions or events). In this respect, the current work aims to enhance the capabilities of large language models through a fine-tuning process that incorporates contextual information and user instructions. The resulting models are expected to be capable of providing personalized recommendations that are aligned with user preferences and desires. More specifically, Generative Pre-trained Transformer 4, a knowledge-based large language model is fine-tuned and turned into a context-aware recommendation system, adapting its suggestions based on user input and specific contextual factors such as location, time of visit, and other relevant parameters. The effectiveness of the proposed approach is evaluated through certain user studies, which ensure an improved user experience and engagement within the museum environment.

**Keywords:** large language models; recommender systems; GPT-4; context awareness; personalization; cultural heritage; museum





## 1. Introduction

The rapid advancement of artificial intelligence (AI) has led to various applications in different domains, including recommendation systems (RSs) [1–4]. These are software tools that assist their users in identifying content to suit their taste from a plethora of available options. They have been widely adopted in areas such as e-commerce and entertainment and play a crucial role in enhancing user experience by providing personalized suggestions, based on individual preferences and needs [5,6]. In the context of museums and cultural spaces, where visitors seek meaningful and engaging experiences, the utilization of advanced RSs can greatly contribute to their enjoyment and overall satisfaction [7,8].

Up until very recently, RSs in museums have been used to either suggest individual exhibits or specific routes to visitors [9], employing various methodologies, such as collaborative filtering, content-based filtering, and stereotyping. In the first case, recommendations are produced based on the content accessed by like-minded users, while, in the second case, the recommended content is similar to that already accessed by the user. In the last case, a "stereotype" is constructed for each visitor, either based on his/her characteristics (age, educational level, etc.) or produced via other means (e.g., answering questionnaires).

The advent of large language models (LLMs) has enhanced RSs by incorporating natural language understanding capabilities. LLMs can be trained on and analyze vast amounts of textual data available in museums, such as exhibit descriptions, relevant literature, and even external sources of information. In this way, they can model the museum space as a whole in a unified way and hopefully serve visitors by providing

more accurate, rich, and context-aware recommendations, in ways similar to experienced human guides.

In light of the above, the current work focuses on the introduction of LLMs—more specifically, Generative Pre-trained Transformer 4 (GPT-4) [10], a knowledge-based LLM— as an RS for museum visitors. GPT-4, an evolution from previous models, possesses exceptional natural language processing capabilities and has the potential to offer valuable recommendations to museum visitors. However, as a language model, GPT-4 lacks inherent context awareness, which is essential in providing accurate and relevant recommendations in museum settings [11].

To address this limitation, the current study proposes the fine-tuning of GPT-4 through the incorporation of contextual information and user instructions during the training process. This fine-tuned version of GPT-4 becomes a context-aware recommendation system, capable of adapting its suggestions based on user input and specific contextual factors such as location, time of visit, and other relevant parameters. By considering context, the system aims to provide personalized recommendations that align with user preferences and desires, ultimately enhancing the overall museum experience.

The primary objective of this contribution is to evaluate the effectiveness of GPT-4 as a context-aware RS in the museum domain. This is achieved by a number of user studies, conducted to measure the system's ability to provide accurate and relevant recommendations, as well as its impact on user engagement and satisfaction within the museum environment. Through these evaluations, the aim is to contribute to advancements in the field of personalized museum recommendations, ultimately benefiting both museum visitors and curators.

This rest of this paper is structured as follows. Section 2 provides an overview of the related work in the field of AI recommendation systems and discusses existing approaches in the cultural domain. Section 3 elaborates on the design of the MAGICAL system. Section 4 describes in detail the design of spaces that were used and the training methodology of the GPT-4 used in MAGICAL. A sample of the training data is also provided. Section 5 discusses the evaluation outcomes along with their implications. Finally, Section 6 concludes the paper by summarizing the findings and outlining future directions for research and implementation.

## 2. Related Work

Recommendation systems have been extensively studied and applied in various domains, including e-commerce, entertainment, and online platforms. However, their application in the context of cultural heritage (CH) and, more specifically, in museums is relatively new and presents unique challenges and opportunities. Moreover, in the last couple of years, the continuous development of GPT has brought about significant changes in various sectors, reshaping interactions with technology. Among the fields profoundly impacted by the advancements in GPT, CH research stands out prominently [12–16].

Agapiou et al. [17] explored the feasibility of using ChatGPT [18], an advanced AI LLM, in remote sensing archaeology. The primary objective was to gain insight into the potential applications of this new language model and understand its capabilities. The authors formulated specific questions based on their scientific expertise and research interests, which were then posed to ChatGPT. The RS model provided responses that appeared satisfactory, although it should be noted that they lacked the comprehensiveness typically achieved through traditional literature review methods.

Grieser et al. [19] focused on offering suggestions to museum visitors according to their past interactions within the museum's physical space, as well as the textual information linked to each item they engaged with. The authors explored an approach to deliver these recommendations using a blend of language modeling techniques, geospatial analysis of the museum layout, and examination of historical sequences of locations visited by previous visitors. In this research, various methods of predicting visitor paths were examined and

compared. The goal was to assess and analyze the effectiveness of these diverse methods for enhancing visitor experience.

Meanwile, Pu et al. [20] introduced a robust recommendation system based on the user's location, aimed at suggesting the most captivating destinations as the user moves around. The system operates by considering both the user's implied preferences and their current physical location, all without necessitating the user to overtly state their preferences or queries. The recommendation process involves identifying places within the physical position circle that also align with the mobile user's implied preferences, as indicated by the virtual preference circle. To estimate the user's implicit preferences, the authors employed a language modeling framework that took into account the user's past visiting patterns. This approach aids in determining the user's interests without requiring explicit input from them.

Also, another work [21] examined established museum chatbots and the platforms used to implement them. Additionally, it presented the outcomes of a systematic evaluation approach applied to both chatbots and platforms and introduced an innovative method for developing intelligent chatbots, specifically designed for museums. The said work prioritized the utilization of graph-based, distributed, and collaborative multi-chatbot conversational AI systems within museum environments. It highlighted the significance of knowledge graphs as a key technology, enabling chatbot users to access a vast amount of information while fulfilling the need for machine-understandable content in conversational AI. Furthermore, the proposed architecture aimed to provide an efficient deployment solution by employing distributed knowledge graphs that could facilitate knowledge sharing among collaborating chatbots, when necessary, as a valuable RS.

Finally, the implementation of an interactive AI RS device, powered by a robust database management system, has been tailored to cater to the distinct needs of various cultural tourists, thereby increasing visitor satisfaction and promoting deeper engagement with the exhibits [22]. Additionally, leveraging data analysis and learning algorithms, AI technology delves into the underlying cultural values, providing valuable insights and inspiration for designing exhibition content, thereby facilitating the digital transformation of museums. Through experimental research, this study demonstrates that the integration of an interactive AI device, based on a database management system, significantly improves the accuracy of the exhibition experience, enhances the effectiveness of voice guidance, streamlines visitor flow management, and incorporates an intelligent RS.

The aforementioned studies provide valuable insight into the development and evaluation of RSs in the museum domain. However, the utilization of GPT-4 as a context-aware RS in museums, as proposed in this paper, represents a novel contribution that combines the strengths of knowledge-based models and fine-tuning techniques to provide personalized and contextually relevant recommendations.

## 3. MAGICAL as a Recommendation System

MAGICAL (Museum AI Guide for Augmenting Cultural Heritage with Intelligent Language model) [12] is a system based on GPT-4 that acts as a digital tour guide in museums. It uses the capabilities of the language model to compose texts and create dialogues with the visitor. For this purpose, the language model must be trained for use in a specific museum or a specific exhibition and be provided with all data necessary to take on the role of the expert. It maintains the style of speech given to it as instruction, can respond in many languages, always remains polite, and avoids any racial, religious, or other discrimination. It can create narratives with real or fictional characters to engage the visitor emotionally, and evaluation has shown that it is able to create recommendations.

In the current work, the objective is to enhance MAGICAL and turn it into a smart RS, providing it with knowledge related to the museum space: the arrangement of objects, their relative position and distance, their topic, the time each visitor is expected to spend at each object, etc. Additional details about each exhibit (descriptions, stories surrounding it (either real or imaginary), dates, information about the creator, etc.) are also supplied to

the system in an effort to transfer the knowledge and experience of someone who knows the space and the exhibits very well. Then, the performance of the enhanced system can be measured based on whether it can correlate visitors' choices with the experience it has accumulated, recommending next steps and keeping the user motivated.

The evaluation of the proposed approach is performed on a virtual (not in the sense of a digital version of an existing museum, but rather a totally fictional one) museum of contemporary art specifically designed for this purpose. The virtual museum considered in the framework of the current work has been labeled as the "Metamorphosis Museum of Modern Art" (MMMA) and is placed in Athens, Greece. The architectural design of its spaces, the distances from various reference points, and the ways to change floors are provided to GPT-4, along with possible places to rest, places to eat and drink, toilets, the museum shop, the cloakroom, and lockers. Moreover, places with digital screens with directions and navigation maps are also provided.

Although there is a wealth of existing literature related to the design of a museum, and a virtual museum in particular [23–28], in the current work, no specific methodology has been selected. The reasons for not adhering to existing design practices can be summarized in the following points: (i) it is desired for the LLM to work and be able to perform outside of any form of known museum design rules; (ii) the specific virtual museum will not be accessible to visitors; (iii) at this stage, the objective is to assess the LLM's ability to learn about the designed museum in detail, so that it can then recommend the next exhibit or route.

Two- and three-dimensional diagrams of MMMA [29] have been designed with the help of the planner5d tool [30], and the same software has been used to create the floor plan [31]. Using the diagram, one can navigate through the museum's space and understand its layout, the relative distances between points, and the ways of proceeding from one floor to another, and can judge whether GPT-4 gives correct navigation instructions to the visitor. The design of the museum is kept minimalistic, as the detailed design of spaces and exhibits is not the goal of this project. Also, from an architectural point of view, it is possible that certain design errors exist in the building, as it is used for evaluation purposes and does not depict an actual place. Therefore, throughout the design of the project, whether on purpose or not, the application of a specific method of setting up the museum, the architectural rules, and reference to existing works of art have been avoided. It was determined that GPT-4 should have no previous connection with the site and the exhibits, only relying on it being trained from scratch.

## 4. Methodology

After laying out the key components of MAGICAL in the previous section, this section proceeds in reasoning about the blueprints of the museum at hand, as well as LLM training, which entails the model's learning of all the museum's spaces and exhibits.

### 4.1. The Design of Space

As the language model cannot provide instructions for places outside of the museum, MMMA's exact location in Athens, Greece is not necessary to be specified. However, its interior layout must be provided to the LLM and it consists of three different levels, as follows:

Ground floor

1.　Entrance Hall: Create an immersive and welcoming experience with interactive installations or digital artworks that introduce visitors to the museum's theme of metamorphosis;
2.　Temporary Exhibitions: Feature rotating exhibitions that highlight contemporary artists, emerging art movements, or thematic explorations. This space can be versatile and adaptable to accommodate different art forms, such as paintings, sculptures, installations, photography, or new media. Two rooms were designed and named Temporary Exhibitions A and Temporary Exhibitions B;

First floor

1.  Modern Painting Gallery: Display a collection of modern paintings from various styles and periods, showcasing the evolution of techniques, styles, and subject matter in modern art;
2.  Sculpture Garden: Sculptures or 3D models of sculptures can be exhibited, allowing visitors to explore different forms, materials, and concepts in modern sculpture;
3.  Video and Digital Art: Create a space dedicated to video installations, digital art, and multimedia presentations. This floor can feature immersive video projections, interactive installations, and digital artwork that push the boundaries of technology and artistic expression;

Second floor

1.  Art Installation: Showcase immersive and large-scale installations that engage the senses and challenge traditional notions of art. These installations can explore themes of transformation, identity, and societal change;
2.  Art Performance Space: Offer a stage for live or recorded performances, including dance, theater, or experimental art forms that blur the boundaries between different art disciplines;
3.  Experimental Gallery: This floor can feature unconventional and boundary-pushing art forms, such as conceptual art, participatory art, or interactive experiences that invite visitors to engage with the artwork.

Figure 1 shows a visual representation of the ground floor with numbered spaces, whose descriptions are summarized in Table 1. Looking at this floor plan, the Entrance and Exit (left in the picture) have been set to be West and, therefore, the long sides of the rectangular building are to the North and South. To train the language model on the design of the specific floor, 30 text sentences or small paragraphs have been used that describe each room individually, the exhibits, and museum operating rules, as well as providing general descriptions, instructions for visitors, distance measurements, etc. Sentences are of varying length. Before these descriptions, there are 12 additional long sentences as general instructions for the whole building, the name and location of the museum, its purpose, and the role of the system itself as a "helpful museum guide who works at MMMA and provides personal guidance and recommendations, according to the visitor needs and the distances involved".

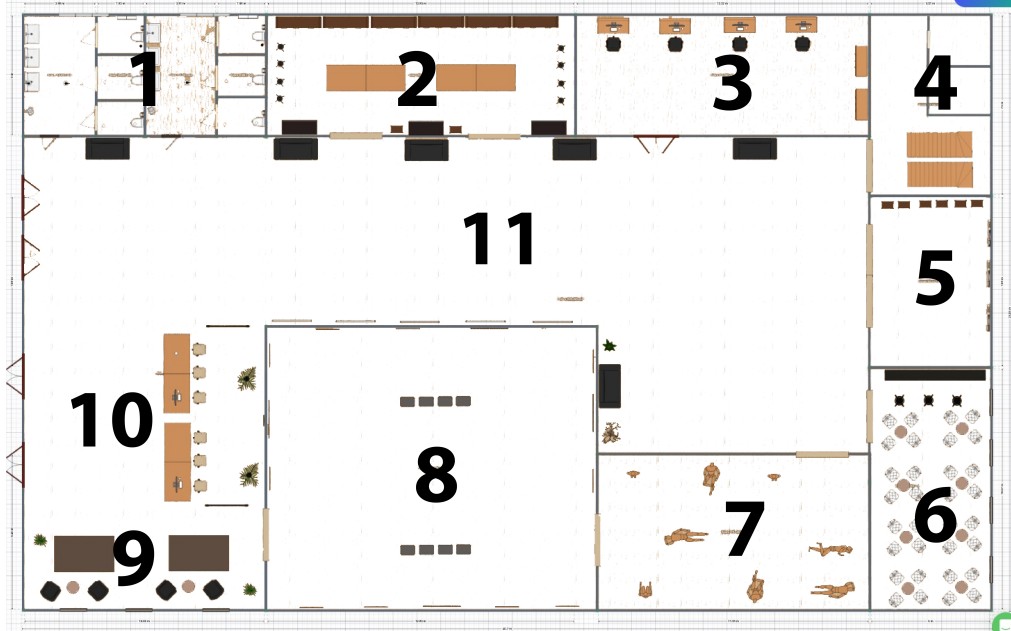

**Figure 1.** Ground floor plan.

**Table 1.** Description of the ground floor plan.

| No | Room Name | Description |
|---|---|---|
| 1 | Toilets | There are two separate rooms for men and women |
| 2 | Cloakroom | In the same room there are lockers for bulky things |
| 3 | Administration | Administration offices. Visitors are not authorized to enter. |
| 4 | Stairs and Elevator | There are two elevators on all floors. |
| 5 | Gift Shop | The museum gift shop. |
| 6 | Cafeteria | Snacks, meals, drinks for visitors. |
| 7 | Temporary Exhibitions A | Wooden human-like sculptures. |
| 8 | Temporary Exhibitions B | Paintings collection. |
| 9 | Lounge | Space for waiting and relaxing. |
| 10 | Reception | The reception desk where visitors buy tickets and get information. |
| 11 | Main corridor | The main walking area. There are interactive screens guiding visitors and sofas for resting. |

GPT-4 has been presented as a multimodal AI system with the ability of recognizing image content and describing it with text [10]. When this feature was publicly released for a short period of time, some short tests were performed in the framework of the current work that did exhibit the ease of the model in describing even complex images. However, as of this writing, this feature is no longer available in the developer API, so it was not possible to test the automatic description of images. This would be a feature that would allow a much faster, and perhaps more accurate, model training, directly from the floor plan images.

Figure 2 offers a visual representation of the first floor, with spaces being numbered and described in Table 2. First floor descriptions consist of 24 sentences that are provided to the model during training. They include information about the architecture (size of rooms, distances, places of entrance and exits, windows, etc.), the exhibits, the creators, and more general instructions on how to move around the space. Personal information and stories about the creators of the exhibits are also given. As discussed before, these are non-existent persons and, thus, their stories are works of fiction. Also, a more general story, of a humorous nature, is provided to examine whether the language model will use it in its dialogues. In this story, an attempt of a famous hacker to break into the server area, where entry to the public is prohibited, to steal museum data is narrated.

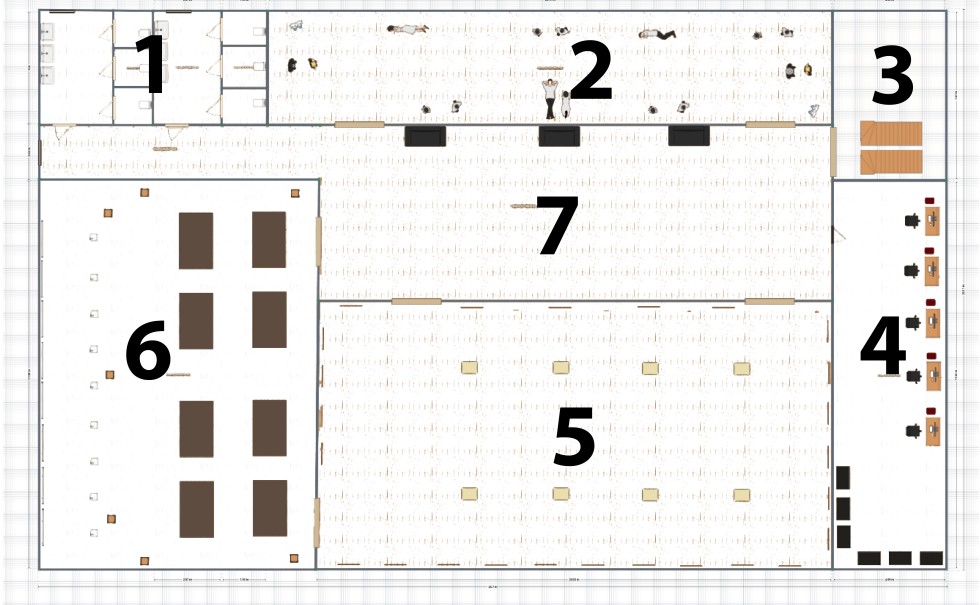

**Figure 2.** First floor plan.

**Table 2.** Description of the first floor plan.

| No | Room Name | Description |
|---|---|---|
| 1 | Toilets | There are two separate rooms for men and women |
| 2 | Sculpture Garden | Sculptures and 3D prints of human-like models with the theme of COVID-19 metamorphosis. |
| 3 | Stairs and Elevator | There are two elevators on all floors. |
| 4 | Data Center | Server room, network, and systems administration. |
| 5 | Modern Painting Gallery | A collection of contemporary paintings. |
| 6 | Video and Digital Art | Space dedicated to video installations, VR applications, and interactive media. |
| 7 | Main Corridor | The main walking area of the first floor. |

Finally, Figure 3 contains a visual representation of the second floor and Table 3 accommodates short descriptions for each area. A total of 41 sentences or paragraphs have been used to describe the second floor. The reason for having more instructional texts here is that 15 contemporary art objects found in the Experimental Gallery room have been detailed. Each of these objects was given a name, a description, and some information about their creator. Accordingly, for the other two main areas of the floor, it was considered that there was an event that was running or starting later on the current day, and descriptions were provided for the titles of the events and their creators.

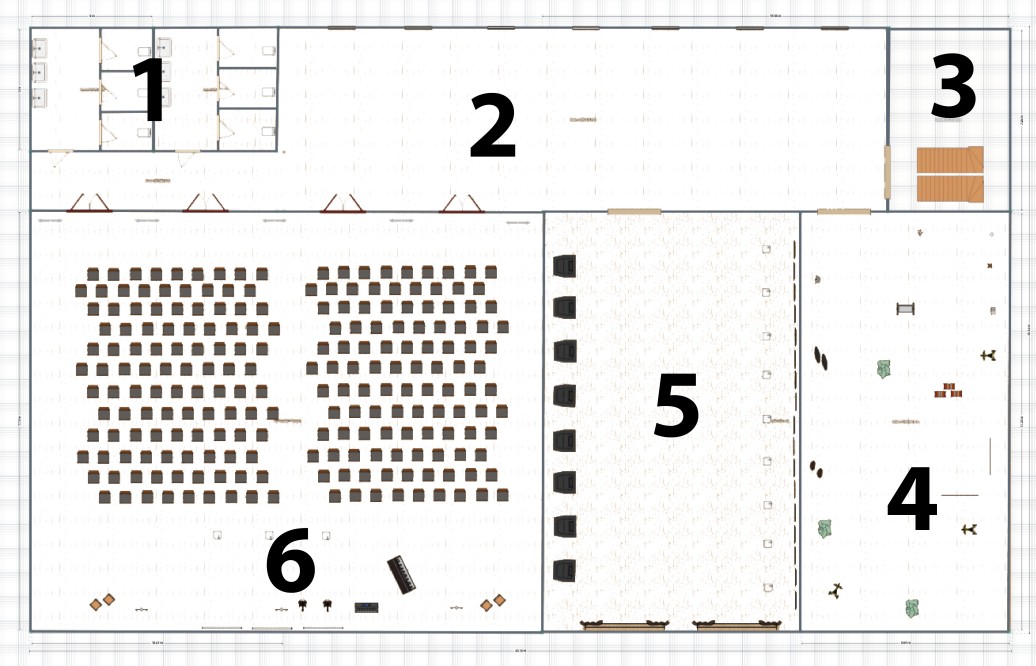

**Figure 3.** Second floor plan.

**Table 3.** Description of the second floor plan.

| No | Room Name | Description |
|---|---|---|
| 1 | Toilets | There are two separate rooms for men and women. |
| 2 | Main Corridor | The main walking area of the second floor. |
| 3 | Stairs and Elevator | There are two elevators on all floors. |
| 4 | Experimental Gallery | A collection of experimental works by several artists. |
| 5 | Art Installations | Room for immersive and large-scale art installations. |
| 6 | Art Performance Space | Stage for live and recorded performances. |

In the Art Installations room, an interactive projection has been placed, called "Digital Realmscape: A Multisensory Journey". In this area, there are screens where users can

interact with the space; at the same time, there are video projectors displaying content throughout the room. The Art Installation is equipped with a smart floor which can sense the presence, location, and number of visitors, and the projected content is changing accordingly. In the Art Performance Space, a concert has been scheduled at the venue, which starts at a specific time on the current day. The system is given details about the concert, artist, and genre and can handle any query. Details about the space, the equipment within, distances, and routes for toilets, stairs, and other areas are also encoded in the system.

*4.2. Model Training*

As is already evident from the preceding section, this particular project entails the existence of a museum with exhibits. However, the involvement of an actual museum with real exhibits has not been considered for the following reasons:

1. Ideally, MAGICAL should be able to function for any museum, even for those not included in the training data of GPT-4;
2. The involvement of an actual museum in this planning phase would have been accompanied by delays due to various practical reasons (meetings with the curators, organizing and ingesting museum literature which may not be in digital form, etc.).

Thus, for the sake of simplicity, speed, and confirmation of the system's own capabilities, it was decided to design an imaginary museum space.

The museum premises, consisting of three floors, were originally designed on paper, where the number and shape of the rooms, the design of the corridors, the stairs, the placement of the elevator, the location of the toilets, the shops, and the reception area were decided upon. Then, these rooms were filled with fictional exhibits and the space was made functional, visitor friendly, and easy to visit without being tedious. A name and a theme were given to the museum; it was geographically placed somewhere in Athens, Greece. The generated, fictional exhibits were given names and descriptions, dates of creation, artists' names, artists' information, and historical information.

In the next design phase, a free online floor planning was employed, which allowed us to publish the design. Within this tool, the drawings were transferred from paper to digital form and details that had not been calculated or planned earlier (the shape and position of windows, the exact location for entrances and exits to each room, objects such as sofas, plant pots, rugs, lamps, etc.) were added, without making the diagram too detailed.

After the creation of the museum space was completed, GPT-4 had to learn about everything a curator and tour guide would know. As a large amount of information had to be ingested by the LLM, a protocol had to be followed so that data were not omitted, repeated, or overlapped. Inconsistencies, ambiguities, and incomplete information had to be avoided.

Figure 4 describes the categorization of the sentences used to train the language model. Sentences about the building, the exhibits, and various narratives were used, as described below. The first sentence defined the role of the model:

"role": "system", "content": "You are a helpful museum guide working in the Metamorphosis Museum of Modern Art (MMMA) in Athens Greece. You are going to be a personal guide and recommender according to the needs of the visitor and the distances involved. Your name is Eva."

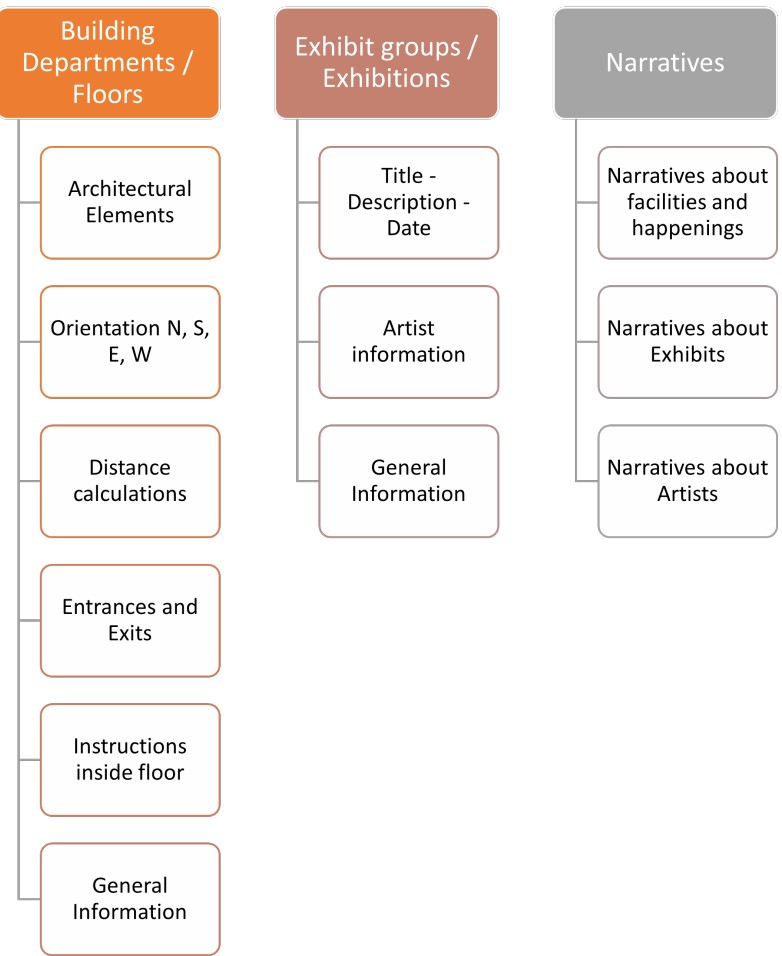

**Figure 4.** GPT-4 training protocol.

It is essential for the training protocol of GPT-4 to be assigned a role before being given any other instruction. The building was initially divided into departments (for easier management) and, in our example, each department was a floor. For each floor, there were instructional sentences that could fit into one of the six defined sub-categories:

- Architectural Elements:
  - "role": "assistant", "content": "There are 3 floors. Ground Floor is the Entrance Hall and Temporary Exhibitions. First Floor is the Modern Painting Gallery, the Sculpture Garden and the Video and Digital Arts Installation. Second Floor is the Art Installation Room (AIR), the Art Performance Space (APS) and the Experimental Gallery.";
  - "role": "assistant", "content": "The stairs and elevators room is rectangular 7.5 m by 5 m. The elevator and stairs room has an open ceiling and the roof of the building can be seen."

- Orientation (North, South, East, West):
  - "role": "assistant", "content": "The MMMMA building is rectangular, with the longest sides facing the North and the South, while the shortest sides face East and West";
  - "role": "assistant", "content": "The toilets are located on the North-West corner of the building. The toilets are identical on all floors, on the same place."

- Distance calculations:

- – "role": "assistant", "content": "The Art Performance Space is the third and last room to find when going on the second floor. Its first entrance is 35 m away from the elevators.";
  – "role": "assistant", "content": "There is a toilet on the second floor, in the exact same area as in every other floor. The entrance for the toilet is 35 m away from the stairs."
- Entrances and Exits:
  – "role": "assistant", "content": "The Experimental Gallery has a single entrance and exit at its north side.";
  – "role": "assistant", "content": "The entrance of the museum is on the West side of the building. There are two big opening doors for the visitors entering the space and two big doors used for the exit."
- Instructions inside floor:
  – "role": "assistant", "content": "As the main corridor of the ground floor is Π-shaped, the edges of the corridor are longer. One of the sofas can be found on the eastern part of the main corridor where there is the biggest opening.";
  – "role": "assistant", "content": "When visitors enter, they are in the Entrance Hall. In front of them there are two reception desks and usually 6 employees are working there, assisting people and selling tickets."
- General Information:
  – "role": "assistant", "content": "Entrance to the museum offices is prohibited for visitors.";
  – "role": "assistant", "content": "MMMA is wheelchair and pushchair accessible."

For administrative reasons, exhibits were grouped into either exhibitions or collections. For each collection, the instructional sentences could fit in one of three sub-categories:

- Title–Description–Date:
  – "role": "assistant", "content": "In the Experimental Gallery visitors can find the exhibit called Silent Grace. It is a life-sized bronze sculpture of a majestic deer, captured mid-stride, exuding a sense of elegance and tranquility. It was crafted in 2012.";
  – "role": "assistant", "content": "In the Experimental Gallery visitors can find the exhibit called Earthen Impressions. They are pottery pieces, each displaying unique textures and organic shapes, celebrating the beauty and artistry found in the marriage of clay and fire. It was crafted in 2017."
- Artist Information:
  – "role": "assistant", "content": "The creator of Digital Realmscape is Lucas Santiago (Age: 32, Country of Birth: Brazil). Lucas Santiago is a visionary multimedia artist known for his innovative approach to merging technology and art. With a background in interactive installations, Lucas has created groundbreaking experiences that blur the boundaries between the digital and physical realms. His passion for creating immersive environments that engage and inspire audiences shines through in the Digital Realmscape: A Multisensory Journey art installation.";
  – "role": "assistant", "content": "In the Experimental Gallery visitors can find the exhibit called Wheels of Freedom. The creator is Alessandro Bianchi (Age: 41, Country of Birth: Italy)."
- General Information:
  – "role": "assistant", "content": "Theatrical and musical performances is something common for the Art Performance Space.";
  – "role": "assistant", "content": "The Art Installation has interactive projectors that can cover the room completely. The sound is high quality surround."

In addition to the above, it would be useful for guests to hear stories related to their experience in the museum. GPT-4 received so much data that it could create narratives on its own, but it could be guided into our own narratives about the museum, the exhibits, and their creators, which would make a visit more interesting. Thus, for evaluation purposes, some sentences were added that told fictional narratives, which, in turn, could be categorized into three sub-categories:

- Narratives about facilities and happenings:
  - "role": "assistant", "content": "Once, somebody tried to enter the Data Center but he was arrested for espionage. Later on it appeared that this man was a famous systems hacker.";
  - "role": "assistant", "content": "The piano in the Art Performance Space is a special handcrafted piano, created by the Greek manufacturer Panos Ioannidis, and it is famous for its quality and sound.";
  - "role": "assistant", "content": "At the Art Performance Space, today at 7:00 p.m. there is a live concert by Markos Konstantakis, a Greek folk musician and singer."

- Narratives about exhibits:
  - "role": "assistant", "content": "Description for the audience of Digital Realmscape is: Embark on a transformative voyage through the 'Digital Realmscape', an immersive art installation that blends technology and imagination. Interact with eight interactive screens that respond to your touch, showcasing mesmerizing visual narratives. The smart floor detects your presence, shaping the experience as you navigate the digital landscapes. Surround yourself with stunning visuals and enveloping sound, as you become an active participant in this captivating multisensory journey.";
  - "role": "assistant", "content": "In the Experimental Gallery visitors can find the exhibit called Pedal Dreams. It is a whimsical baby bicycle, adorned with colorful patterns and symbols, evoking a sense of nostalgia and the simplicity of childhood joys. This exhibit was previously on display at the Guggenheim Museum, where there was a failed attempt to steal it."

- Narratives about artists:
  - "role": "assistant", "content": "Inside the Video and Digital Arts, the interactive videos were created by a Greek artist called John Aliprantis. John is 35 years old, he comes from Paros Greece and this is his first public exhibition. John stated in an interview in the past few days that he is particularly happy that his works are being presented at the MMMA.";
  - "role": "assistant", "content": "Inside the Video and Digital Arts, the music is changing automatically according to the visitors' behavior in the room. The music is composed by George Trichopoulos, a Greek musician, aged 51. George has gained 5 Grammy awards in his career."

In total, approximately 170 sentences were used to guide the language model in relation to the museum and its exhibits. Every time a user initiated dialogue with the system, all data were ingested into GPT-4. The system then stepped into its role and supported the dialogue very well. If the answer to a question did not seem correct, it further evaluated whether the language model had given wrong instructions or no instructions at all (in relation to the question).

## 5. Primary Evaluation

The LLM would require a lot of data about the museum's operation in order to function as a RS. OpenAI's language models process textual information in the form of tokens [32], which are data units that roughly correspond to one syllable of a word. This mapping is not always accurate, as popular words may "occupy" one token, while infrequent words may be assigned a token even on the letter level. In earlier versions of GPT, there was a limit of using/sending up to 4096 tokens per conversation. Even though this number is

considered to be sufficient for small-scale applications and simple experiments, in the case of MAGICAL, more than 18,000 tokens are estimated to be exchanged during each dialog. In cases of errors, new sentences with instructions to the model have to be formed and the number of tokens needs to be increased further.

The language model exhibited satisfactory results in the evaluation experiments, working as expected. It gave wrong answers sometimes, but this behavior is attributed to contradictions or deficiencies in the instructions. The response language did not present any syntactical or grammatical errors. Even though English was the language most commonly used, it is very easy to switch to another language. In this context, Greek was used as the second language during evaluation, but the model presented some small weaknesses in this case, predominately in the use of the language. There were minor syntactical and semantic errors, as the model actually works in English and then translates to other languages.

A particularly positive impression is the fact that the model can answer questions for which it has not been previously trained. Figure 5 displays the graphical interface of the system under development, while the quoted text below presents the dialogue in one of the evaluation experiments. The system correctly answers questions regarding directions to the area and then it is asked whether the museum's gift shop is expensive. The returned answer is completely logical and correct, despite the fact that it was not included in the training data. The previous question related to this dialogue was "What metamorphosis means?". Here again, the system had not been given any relevant instructions and responded on its own, contextually.

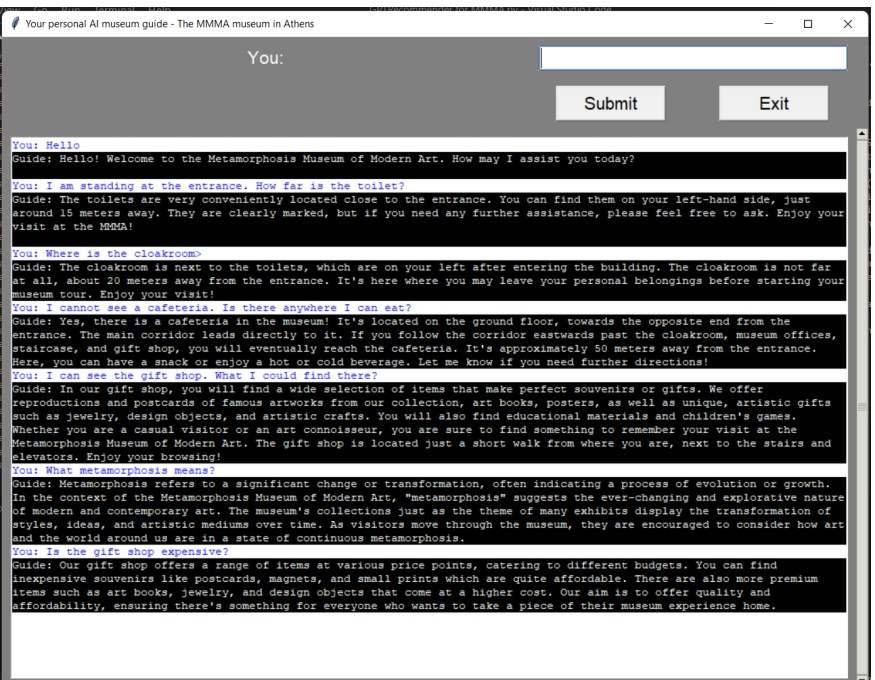

**Figure 5.** GPT-4 answering questions, including ones it had not been trained upon.

> You: Hello
>
> Guide: Hello! Welcome to the Metamorphosis Museum of Modern Art. How may I assist you today?
>
> You: I am standing at the entrance, How far is the toilet?
>
> Guide: The toilets are very conveniently located close to the entrance. You can find them on your left hand side, just around 15 m away. They are clearly marked, but if you need any further assistance, please feel free to ask. Enjoy your visit at the MMMA!
>
> You: Where is the cloakroom?

Guide: The cloakroom is next to the toilets, which are on your left after entering the building. The cloakroom is not far at all, about 20 m away from the entrance. It's here where you may leave your personal belongings before starting your museum tour. Enjoy your visit!

You: I cannot see a cafeteria. Is there anywhere I can eat?

Guide: Yes, there is a cafeteria in the museum! It's located on the ground floor, towards the opposite end from the entrance. The main corridor leads directly to it. If you follow the corridor eastwards past the cloakroom, museum offices, staircase, and gift shop, you will eventually reach the cafeteria. It's approximately 50 m away from the entrance. Here, you can have a snack or enjoy a hot or cold beverage. Let me know if you need further directions!

You: I can see the gift shop. What I could find there?

Guide: In our gift shop, you will find a wide selection of items that make perfect souvenirs or gifts. We offer reproductions and postcards of famous artworks from our collection, art books, posters, as well as unique, artistic gifts such as jewelry, design objects, and artistic crafts. You will also find educational materials and children's games. Whether you are a casual visitor or an art connoisseur, you are sure to find something to remember your visit at the Metamorphosis Museum of Modern Art. The gift shop is located just a short walk from where you are, next to the stairs and elevators. Enjoy your browsing!

You: What metamorphosis means?

Guide: Metamorphosis refers to a significant change or transformation, often indicating a process of evolution or growth. In the context of the Metamorphosis Museum of Modern Art, "metamorphosis" suggests the ever-changing and explorative nature of modern and contemporary art. The museum's collections just as the theme of many exhibits display the transformation of styles, ideas, and artistic medium over time. As visitors move through the museum, they are encouraged to consider how art and the world around us are in a state of continuous metamorphosis.

You: Is the gift shop expensive?

Guide: Our gift shop offers a range of items at various price points, catering to different budgets. You can find inexpensive souvenirs like postcards, magnets, and small prints which are quite affordable. There are also more premium items such as art books, jewelry, and design objects that come at a higher cost. Our aim is to offer quality and affordability, ensuring there's something for everyone who wants to take a piece of their museum experience home.

Another positive aspect of the model is that it could combine data to produce insights. In one example, assuming that the visitor was on the second floor and inside a room, the question "how far is the exit of the museum?" was asked. The instructions received by the system during its training concerned each floor separately. The model added the distances from the visitor's point to the elevator, calculated the time to the first floor, and added the distance from the elevator to the exit on the ground floor before giving the correct answer.

Using the protocol proposed in [33], some complete dialogues with the system have been recorded in the Supplementary Text File "Conversation with MAGICAL". As general additional remarks on these dialogues, we can observe the following:

- Accidental spelling mistakes by the user do not affect the dialog. For example, the authors ask the system "What Metamorphisis stands for?" instead of using the word metamorphosis. This does not confuse the system and does not affect its response. Even if there are multiple spelling errors in a sentence, the system continues to respond. In the final dialogue provided, all questions contain grammar and syntax errors to test the robustness of the system. Additionally, the dialogue switches from English

to French, Greek, and, finally, Italian, always with misspellings, but this does not seem to affect system performance;

- The system never provides a response that contradicts the instructions it has received. Conversely, when it has not been instructed about something being asked, it connects information and prior knowledge and then provides answers in a completely logical way. For example, in one dialogue it was asked: "Is it allowed to capture video inside the AIR?". The model has been instructed about the meaning of the initials AIR, which is just a room in the museum, but it was never given any instruction about whether filming or photo-shooting is allowed inside the museum. The answer it came up with is, "Yes, you are allowed to take photos and videos inside the Art Installation Room (AIR) for personal use only, not for commercial purposes. However, we strongly advise our visitors to experience and enjoy the exhibit rather than focusing on capturing it on their devices. Also, please remember to be respectful to other visitors who are also experiencing the exhibit. Thank you for your understanding and enjoy your visit!". Thus, the model not only manages to answer questions that it theoretically does not know, but also reaches the point of giving positive advice for a better visit experience.

Model training cost for a particular application is highly dependent on the employed GPT version. The newest version (GPT-4, as of this writing) exhibits a tenfold improvement in the required cost, while training the model for a new domain is relatively easy. As described above, a methodology has been created for training the model so that it can be easy to transfer the application to any space, cultural or not. During the first tests, authors anticipated the functionality of the application, its robustness to errors, and its ability to function as a real RS. User evaluation in an actual museum space is within the scope of future plans, in order to evaluate user reactions and estimate user experience.

## 6. Conclusions

Large language models are still unexplored for use in areas such as cultural heritage. They exhibit rapid development and even greater possibilities. GPT-4 is a text composition system that shows great flexibility. Although it is trained on a huge amount of data, it can be guided with great ease and take on various roles for applications in the fields of cultural heritage, education, tourism, video games, etc. In the experiments, it was demonstrated that it could be transformed into a very informative recommendation system, making natural language dialogues and even answering complex questions. This project is the continuation of a set of research applications related to GPT-4 [12] and, as a following step, voice command integration will be added. Communication with MAGICAL is carried out in natural language (oral form), without the use of screens. Once the application reaches its final form, it is expected to be tested in a real museum, for evaluation using visitors.

**Supplementary Materials:** The following supporting information can be downloaded at: https://www.mdpi.com/article/10.3390/electronics12183829/s1, Supplementary Text File: Conversation with MAGICAL.

**Author Contributions:** Conceptualization, G.T.; methodology, G.T.; software, G.T.; validation, G.T., M.K. and G.A.; formal analysis, G.T.; investigation, G.T.; resources, G.T.; data curation, G.T. and G.C.; writing—original draft preparation, G.T. and M.K.; writing—review and editing, G.T., M.K. and G.A.; visualization, G.T.; supervision, G.C.; project administration, G.T. and M.K.; funding acquisition, M.K. All authors have read and agreed to the published version of the manuscript.

**Funding:** This research received no external funding.

**Data Availability Statement:** Not applicable.

**Conflicts of Interest:** The authors declare no conflict of interest.

## Abbreviations

The following abbreviations are used in this manuscript:

| | |
|---|---|
| AI | Artificial Intelligence |
| API | Application Programming Interface |
| CH | Cultural Heritage |
| GPT | Generative Pre-trained Transformer |
| LLM | Large Language Model |
| MAGICAL | Museum AI Guide for Augmenting Cultural Heritage with Intelligent Language model |
| MMMA | Metamorphosis Museum of Modern Art |
| RS | Recommendation System |
| VR | Virtual Reality |

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
