# Peer review of "Large Language Models as Recommendation Systems in Museums"

_electronics, doi:10.3390/electronics12183829_

Round 1

Reviewer 1 Report

The paper delves into an innovative use of the GPT-4 language model in a virtual museum, as a recommender system. The authors put much effort into preparing this work; the paper is quite flawless and presents the ideas and experimental results in an engaging manner.

The paper needs to focus more on the topic of recommendation systems; in its current shape it does not tell the reader much about what a recommender system is, and what other types of RSs there are (so as to provide the basis for the reader to see how innovative the use of GPT-4 is in this context). I suggest adding a concise paragraph explaining this. 

In addition to this, I'd be personally interested to see more of the model's outcomes and interactions with users, even mistakes or errors. But that's just preference, not criticism of the work. 

The overall level of English is adequate, correct and comprehensive. There are a few details to be corrected before the paper could be published, such as:

- the use of the informal abbreviation "can't" in the Abstract

-"context-ware' instead of "...-aware", in the Abstract

-"overview of related work..." - "overview of the related..." p. 2

-a misplaced comma in p. 3 after "fictional one)"

- p.6 - "publicly release for" - "publicly released for"

Author Response

Dear Editor,

We would like to thank you for giving us the opportunity to revise and resubmit our manuscript titled “Large Language Models as Recommendation Systems in Museums”.  We found the reviewers’ comments to be helpful and constructive and we have carefully considered them in an effort to substantially improve the quality of our work. In this submission, we have included a response to each reviewer, where every comment and suggestion made is addressed.

Once again, thank you again for considering our revised manuscript.

On behalf of the authors of the submission electronics-2541277,

Sincerely,

Georgios Trichopoulos,

Intelligent Interaction Research Group,

Department of Cultural Technology and Communication

University of the Aegean,

Mytilene, Greece

Reviewer 2 Report

electronics-2541277-peer-review-v1

Review of:  Large Language Models as Recommendation Systems in Museums

This is a very topical and interesting paper.

I have some concerns that will need to be addressed in a major review of the manuscript.

1) While appreciate the flow of the paper, there needs to be Methodology section that sets out clearly what approaches the authors took. At this point, some is this is embedded in the text of section 3, but not in a very transparent way. If I understand it correctly, the author’s designed a fictitious museum space with a planner tool and then used Chat GPT to design the exhibit that filled that space. This is not that clearly expressed.

2)         Who designed the ground floor plan shown in figure 1. The authors via a planner tool or Chat GPT? It seems from the text that this layout was fixed and fed into ChatGPT incl. contents of the rooms. For ex. Temp Exhibit A and B give guidance to ChatGPT what should be shown there. Is that appropriate? 

3) Line 192 ff. The same applies to the other floors. It seems that considerable design guidance was provided that may have constrained ChatGPT in developing its own concepts (see https://www.preprints.org/manuscript/202307.1523/v1 ) This section provides even more guidance and detail.

When I finally arrived at the snippet of conversation described in Figure 4 it became obvious that the whole focus of the paper was to merely act as  guidance and Q&A system for visitor rather than as a planning tool for museum curators. That needs to be discussed in much more detail and also be flagged vey early on.

Line 255          “A methodology has been created for training the model” Where is that methodology? Is that the provision of a floor plan and the contents of each room? 

I consider much of the description too cursory to be of much value at this point and encourage the authors to add more depth to their description.

It is also not quite clear how and by whom the model was evaluated. Did the authors bring an external museum communicator or educator to look at the responses?

Also, Fig 4 shows answers without prior training. What was the quality of the answers AFTER training?

Finally, I am missing the data set used for the paper. Ideally, the conversation should be made accessible. See the following protocol (https://www.preprints.org/manuscript/202307.2035/v1) (which, granted) seems to be more recent than the manuscript. Still, the authors should follow this in their methodology).

Author Response

(The authors gave the same response as above.)

Reviewer 3 Report

Firstly, congrats to the authors for your valuable research. This topic is interesting, and the manuscript is simply an understandable and well-structured presentation. However, I suggest several points in this manuscript should be substantially improved for publication.

More related work is needed to fully address the research limitation.

Although the experimental results are impressive, the proposed method is not described clearly. I suggest the authors describe the method more clearly and make a simple example to explain how the model deploys. If possible, it would be more useful to provide an illustrative example with limited data to explain the steps one by one of the proposed method.

The paper is comprised of many grammar mistakes. I suggest the authors carefully rewrite the paper or use the proofreading services before re-submission

The similarity checking is high (26%). Please consider rephrasing all similar sentences. I attached the checking results for reference (checked by Turnitin service)

Author Response

(The authors gave the same response as above.)

Round 2

Reviewer 2 Report

I have carefully read the revised versiona nd would I would like to congratulate the authors on the revision of the paper. They have adequately addressed all concerns.

One minor issue emerged in the revision.
I am confused why the design and description of the rooms is a section that comes before the Methodology. Surely, the room design is part of the methodology and should be subpoint of this?

So, 

4. Methododlogy Insert at line 176

intro sentence of two

4.1. Design of Space

4.2. Methodology of Evaluation

then lines 

Author Response

Dear Reviewer,

We would like to thank you for giving us the opportunity to revise and resubmit our manuscript titled “Large Language Models as Recommendation Systems in Museums” for a second time.  In this revision round, we found the reviewers’ comments helpful and naturally we have considered them in an effort to further enhance the quality of our work. As in the previous revision round, in this resubmission, we have included a response to each reviewer, where every comment and suggestion made is addressed.

Once again, thank you again for considering our revised manuscript.

On behalf of the authors of the submission electronics-2541277,

Sincerely,

Georgios Trichopoulos,

Intelligent Interaction Research Group,

Department of Cultural Technology and Communication

University of the Aegean,

Mytilene, Greece

Reviewer 3 Report

I think this revised version is good enough to consider for publication

Author Response

(The authors gave the same response as above.)
